# LLM-enhanced Self-training for Cross-domain Constituency Parsing

**Jianling Li**[1]    **Meishan Zhang**[2]    **Peiming Guo**[3]    **Min Zhang**[2]    **Yue Zhang**[3*]

[1]School of New Media and Communication, Tianjin University, China
[2]Institute of Computing and Intelligence, Harbin Institute of Technology (Shenzhen), China
[3]School of Engineering, Westlake University, China
jianlingl@tju.edu.cn, mason.zms@gmail.com, guopeiming.gpm@gmail.com
zhangmin2021@hit.edu.cn, yue.zhang@wias.org.cn

## Abstract

Self-training has proven to be an effective approach for cross-domain tasks, and in this study, we explore its application to cross-domain constituency parsing. Traditional self-training methods rely on limited and potentially low-quality raw corpora. To overcome this limitation, we propose enhancing self-training with the large language model (LLM) to generate domain-specific raw corpora iteratively. For the constituency parsing, we introduce grammar rules that guide the LLM in generating raw corpora and establish criteria for selecting pseudo instances. Our experimental results demonstrate that self-training for constituency parsing, equipped with an LLM, outperforms traditional methods regardless of the LLM's performance. Moreover, the combination of grammar rules and confidence criteria for pseudo-data selection yields the highest performance in the cross-domain constituency parsing [1].

## 1 Introduction

Constituency parsing, a fundamental task in natural language processing (NLP), has achieved remarkable progress on in-domain benchmarks (Liu and Zhang, 2017; Gaddy et al., 2018; Kitaev and Klein, 2018; Kitaev et al., 2019; Cui et al., 2022), indicating the growing competence of parsers in capturing the underlying syntactic structures. However, open-domain constituency parsing is notably challenging (Fried et al., 2019; Yang et al., 2022). In diverse, open-domain scenarios, constituency parsing faces complexities beyond the well-defined task. Addressing these challenges is crucial for its broader real-world NLP applications.

To address the issue of domain shift, self-training-based unsupervised domain adaptation has emerged as a promising approach (Yu et al., 2015; Sachan and Xing, 2018; He et al., 2019; Rotman

---

and Reichart, 2019; Ramponi and Plank, 2020; Ye et al., 2020; Wang et al., 2021). This method utilizes a source domain model to automatically label a large-scale raw corpus from the target domain during each iteration. High-confidence pseudo data is then selected as additional training data to improve target domain performance. However, the quality and quantity of raw corpus cannot always be guaranteed for low-resource domains (Steedman et al., 2003; Qiu et al., 2014; Peng et al., 2021), which limits the use of self-training approaches. Traditional methods struggle to construct fine-grained sentences that facilitate knowledge transfer. The Large Language Model (LLM), with its powerful generative capabilities, can serve as a potential solution to the challenge of the raw corpus quantity and quality for the target domain (as shown in Figure 1). It's important to note that our experiments revealed that LLMs exhibit limited performance for constituency parsing.

To tackle the challenges of LLMs' flexibility and hallucination problems (Bang et al., 2023; Manakul et al., 2023) in generating sentences, we employ grammar rules as instructions for LLMs to generate target domain sentences. Grammar rules have proven effective in cross-domain data generation (Wang et al., 2023) and are closely related to constituency parsing (Yang et al., 2022). By incorporating grammar rules, we aim to generate a large volume of high-quality raw sentences highly relevant to the cross-domain constituency parsing task. Furthermore, we dynamically embed LLMs into the iterative process of self-training, enhancing their adaptability and flexibility.

In each iteration of self-training, we 1) use LLM with grammar rules extracted from currently available training instances to generate target domain raw corpus; 2) train a constituency parser with this data; 3) parse the raw corpus using the trained parser; and 4) select high-quality pseudo data from the parsed trees as additional training instances for

---

*Corresponding author.

[1]We have made our code publicly available at https://github.com/jianlingl/LLM_ST_ConstParsing

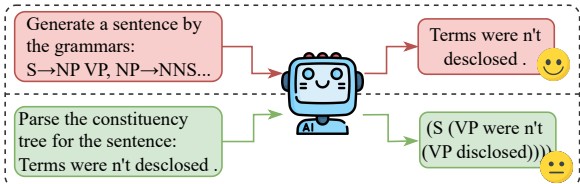

Figure 1: the capabilities of LLMs. Despite encountering parsing challenges such as missing elements, LLMs can generate reliable sentences using grammar rules.

the next iteration. For data selection, we employ multiple criteria such as token-based, confidence-based, and grammar-rule-based selection.

Experimental results demonstrate the superior constituency parsing performance of our LLM-enhanced self-training approach. The raw corpus generated by LLM proves quite effective for the cross-domain self-training tasks, and combining grammar-rule-based and confidence-based selection of pseudo data performs best. This criterion considers both structural information and ensures reliability. Additionally, open-source LLMs can effectively replace closed-source LLMs, achieving comparable performance in our methods. Throughout the self-training iterations, updated grammar rules and selected pseudo data progressively move closer to the target domain.

To the best of our knowledge, we are the first to dynamically incorporate LLMs into the self-training for constituency parsing. Our model utilize LLMs to create a fine-grained raw corpus that gradually adapts from the source to the target domain. We also introduce an optimal pseudo-data selection criterion, combining grammar-rule-based and confidence-based criteria. This selection effectively contributes to our LLM-enhanced self-training method for cross-domain constituency parsing. Furthermore, we observed that LLMs underperform in few-shot settings compared to both our baselines and our proposed method.

## 2 Related Work

**Cross-domain Constituency Parsing** Constituency parsing, a long-standing traditional NLP task, has evolved over the years (Collins, 1997; McClosky et al., 2006; Zhu et al., 2013; Dyer et al., 2016; Stern et al., 2017; Gaddy et al., 2018). Supervised constituency parsing (Liu and Zhang, 2017; Kitaev and Klein, 2018; Kitaev et al., 2019; Cui et al., 2022) has achieved remarkable performance, but it requires massive manual efforts to annotate the treebank (Marcus et al., 1993; Xue et al., 2005; Seddah et al., 2013). In contrast, unsupervised or weakly supervised open-domain parsing remains a challenging research task (Yang et al., 2022). There are limited studies on cross-domain constituency parsing (McClosky et al., 2010; Fried et al., 2019).

**Self-training** Researchers (Wang et al., 2015; Yang et al., 2022) have attempted to directly transfer source domain parsers to target domains. However, such direct transfer approaches are insufficient for target domains with large gaps compared to the source domain. Self-training (Yarowsky, 1995; McClosky et al., 2006; Yu et al., 2015; Ramponi and Plank, 2020; Guo et al., 2022) is a simple and early bootstrapping approach for domain adaptation, applied to tasks such as classification Dong and Schäfer (2011); Ye et al. (2020), sequence labeling Wang et al. (2021, 2020), dependency parsing Yu et al. (2015); Rotman and Reichart (2019); Guo et al. (2022), sequence generation He et al. (2019), and QA Sachan and Xing (2018) and so on. However, self-training has not been applied to cross-domain constituency parsing.

**LLMs' Parsing** LLMs have been successfully applied to various NLP tasks (Brown et al., 2020; He et al., 2023; Mysore et al., 2023; Zhang et al., 2023; Pangakis et al., 2023; Ashok and Lipton, 2023). While LLMs exhibit limitations in structured extraction tasks (Qin et al., 2023), such as cross-domain constituency parsing, their generative capabilities can be leveraged to provide a raw corpus for self-training constituency parsing.

To address the issue of hallucinations in LLMs, we propose employing grammar rules to regulate the structure of generated sentences. We also provide a small set of domain-specific sentences as style references and length limits to further guide the LLM generation process. Prior research Yang et al. (2022) has shown a close relationship between grammar rules and constituency parser performance, while Wang et al. (2023) successfully used grammar rules to generate cross-domain data. Our work uniquely integrates LLMs into self-training iterations, generating fine-grained raw corpus based on the grammar rules, making the process more flexible and controlled.

**Pseudo-data Selection Strategy** When applying the self-training method to different tasks, researchers typically develop task-specific criteria

for pseudo-data selection, taking into account the unique characteristics of each task. For example, Dong and Schäfer (2011) use an ensemble learning model for reliable newly labeled data selection in classification. Wang et al. (2020) show that sampling strategies may vary across datasets, and Jiang et al. (2021) propose using out-of-vocabulary numbers to measure source-target domain distance in Chinese word segmentation and POS tagging. In this work, we propose adopting grammar rules as a selection criterion and combining it with confidence-based selection. This method ensures the pseudo-data contains rich structural information and exhibits high reliability, ultimately improving cross-domain constituency parsing performance.

## 3 Method

### 3.1 Baseline Model

We employ the Berkeley Neural Parser (Kitaev and Klein, 2018) as the foundation of our method. This parser is a chart-based approach that adopts a self-attentive encoder and a chart-based decoder, utilizing pre-trained embeddings as input to enhance the parsing process.

Due to the integration of pre-trained language models, the Berkeley Neural Parser inherently possesses cross-domain constituency parsing capabilities. This enables the parser to be trained on the source domain and then directly applied to parse trees for the target domain. We use this direct model transfer as a baseline for comparison.

Additionally, we compare the cross-domain parsing performance of the smaller model (e.g., Berkeley Neural Parser) with that of the LLMs constituency parsing. Specifically, we provide the Large Language Model with a few parse trees from the source domain as prompts and ask it to generate parse trees for the target domain. This comparison further helps us understand the strengths and weaknesses of both large and small models when applied to constituency parsing tasks.

### 3.2 Vanilla Self Training

In this section, we introduce a vanilla self-training method for cross-domain constituency parsing, which has been investigated in other tasks. Please note that we refer to the standard self-training method as vanilla self-training. The primary goal of self-training is to generate high-quality training instances for the target domain, subsequently using these instances to train the target domain model.

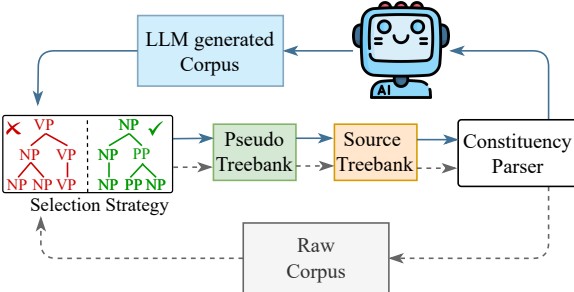

Figure 2: Vanilla and LLM-enhanced self-training frameworks for cross-domain constituency parsing. The former is represented by the lower gray dashed line loop, while the latter is depicted by the upper blue solid loop.

The vanilla self-training-based cross-domain constituency parsing is an iterative process aimed at training a target parser.

Specifically, in each iteration of the vanilla approach, three main steps are conducted: 1) Training the parser: We train the Berkeley Neural Parser using the source domain constituency treebank. 2) Parsing raw corpus: We apply the trained model to parse the raw text from the target domain, generating parse trees that serve as candidate pseudo trees for the next step. 3) Selecting pseudo-data: We select high-confidence pseudo trees to serve as additional training instances, which are then used to enhance the model performance on the target domain. By repeating these steps iteratively, the self-training method adapts the parser to the target domain, leveraging both the source annotated treebank and high-quality pseudo trees generated throughout the process.

### 3.3 LLM-enhanced Self-training

To improve the quality and quantity of raw corpus used on vanilla self-training, we propose to integrate LLM into the self-training iteration as shown in Figure 2. We dynamically embed the LLM as a crucial component in our iterative self-training process. In each iteration, we utilize the LLM to generate raw corpus for the target domain, based on the updated treebank from the previous step. Following the detailed LLM-enhanced self-training constituency parsing algorithm 1, our method requires an annotated treebank from the source domain, as well as a small number of sentences from the target domain. The **G**rammar **R**ules ($GRs$) are extracted from the treebank and play a crucial role in guiding the LLMs generation of raw corpus for target domain.

**Algorithm 1:** LLM-enhanced Self-training

**Input:** Source Constituency Treebank $S$
Treebank Grammar Rules $GRs$
Small Target Domain Raw Corpus $R$

**Output:** Target Constituency Parser $M$
Pseudo Target Treebank $\widehat{D}$

1   $\widehat{D} = \{\}$
2   **foreach** *i in [1, 2, . . . ] until convergence* **do**
3     LLM Generating: $\widehat{R} = GPT(GRs, R)$
4     Parser Training: $M = train(S \cup \widehat{D})$
5     Domain Parsing: $D = parse(M, \widehat{R})$
6     Trees Selection: $\widehat{D} = topK(D)$
7     Treebank Update: $S = S \cup \widehat{D}$
8     GRs Extraction: $GRs = extract(S)$
9   **end**

We divide the LLM-enhanced self-training constituency parsing into six detailed steps on each iteration: **1) LLM Generating**: We first leverage the Large Language Model to produce a raw corpus $\widehat{R}$ for the target domain, based on $GRs$ extracted from the currently available treebank and a few sample sentences ($R$) from the target domain. **2) Parser Training**: Next, we train a constituency parser using the source treebank $S$ and the selected pseudo trees $\widehat{D}$ for the target domain. During the initial step, the pseudo treebank isempty ($\widehat{D} = \{\}$), and the parser is trained solely on the source domain data. **3) Domain Parsing**: We apply the trained parser to parse the generated raw corpus $\widehat{R}$, resulting in a set of candidate parse trees $D$. **4) Trees Selection**: From the generated parse trees $D$, we select a subset of high-quality parse trees to form the pseudo treebank $\widehat{D}$. The selection criteria detailed in subsection 3.4. **5) Treebank Update**: We update the source treebank $S$ by adding the selected pseudo treebank $\widehat{D}$ to it, effectively increasing the diversity of training data for the target domain. **6) GRs Extraction**: We extract grammar rules $GRs$ from the updated treebank $S$, which will guide the LLM to generate more Informative raw corpus for the target domain in the next iteration. The LLM-enhanced self-training process is iteratively continued until convergence. The final output of the algorithm is a trained target constituency parser $M$ and a pseudo treebank $\widehat{D}$ for the target domain. This approach leverages the Large Language Model generated raw corpus, which replaces the vanilla ready-made text, enhancing the adaptation

process and improving the parser's performance on the target domain throughout the iterations.

### 3.4 Instanecs Selection Criteria

To align with the goal of the LLM-generated raw corpus, which aims to incorporate as much structural information as possible by grammar rules to improve constituency parser performance, we propose a grammar-rule-based selection criterion for pseudo-data. Unlike previous self-training selection criteria that focus solely on the task, this criterion considers both the task and the characteristics of the LLM-generated corpus, ensuring that the selected pseudo-data is appropriate for cross-domain parsing using self-training.

In particular, LLMs generate sentences that closely resemble the target domain using grammar rules. Our selection criterion, on the other hand, employs grammar rules to choose pseudo-data that is as relevant as possible to the source domain, reducing potential transfer failures due to large biases and gaps. The feasibility of grammar-rule-based selection criteria is also supported by Yang et al. (2022) and Wang et al. (2023).

However, directly measuring the distribution disparity between a training set and a candidate instance can be challenging. We provide a high-level inspection of how to evaluate the distance between a large set and an individual instance. Given the source set, denoted as $S$, and the candidate instance, denoted as $c \in C$ (candidate set), we describe the distance computation between $c$ and $S$ using equation (1):

$$D(c, S) = JS(S, S \cup \{c\}) \qquad (1)$$

$$instances = topK \arg\min_{c \in C} D(c, S) \qquad (2)$$

We then pick the topK candidates closest to the source domain set as additional training instances in the self-training process. This approach ensures that the most relevant instances are selected, enhancing the model's gradual adaptation to the target domain.

The distance computation can be performed at either the token level or the grammar rule level by adjusting the set to represent token distribution or grammar rule distribution, respectively. The grammar rules we use include both terminal and non-terminal rules. Our instance selection process involves three levels of criteria: token, confidence,

and grammar rule. We also combine the two best-performing criteria, namely confidence-based selection and grammar-rule-based selection, resulting in a more effective criterion for identifying high-quality instances for adaptation to the target domain.

- Token-based criteria prioritize trees with word distributions closer to the source treebank (using the JS distance as in equation (2)).

- Conf-based criteria focus on the scores of labeled phrases provided by the model to identify high-confidence pseudo trees.

- GRs-based criteria favor instances with similar structures by calculating the JS distance between the candidate tree and the source grammar rules set.

- GRsConf-based criteria select high-confidence instances among candidates with high scores on the grammar rule, considering both structural information and data reliability.

### 3.5 LLM prompt

To generate sentences that encompass comprehensive structural information and closely resemble the target domain sentence style, we introduce a LLM prompt that integrates grammar rules and target domain examples. During the generation, we need to prepare the following parameter: 1) $N$ grammar rules extracted from the treebank, 2) $M$ sampled sentences from the target domain, and 3) length constraints $L_1 \sim L_2$ for the generated sentence to ensure they are neither too short nor too long.

Through preliminary experiments, we have found a direct correlation between the number of grammar rules and the length of LLM generated sentences. Therefore, we determine the value of $N$ by sampling from the distribution of treebank sentence lengths from which we extract the grammar rules. Note that the grammar rules are directly extracted from the constituent tree, where the parent node corresponds to the left hand of the grammar rule, and all child nodes correspond to the right tail side. For instance, if the treebank is the source domain data PTB, we introduce a Gaussian distribution for the averge length, denoted as $N = \mathcal{N}(avg\_len, 6)$ to obtain $N$ grammar rules.

The number of target domain sentences to be extracted is customizable, but due to resource constraints and minimal performance differences, we

---

**LLM Prompt**

As a language assistant, you excel at creating sentences of a specific length while adherating to $N$ grammar rules provided above. Please consider the $M$ examples, generate one sentence of $L1 \sim L2$ words:

GRs: S→PP NP VP,   PP→IN NP,   VP→VP NP, ...

Snts: 1. History brings inspiration to peace . 2. It is the jack of all trades and master of none . 3. ... ...

Figure 3: LLM prompt example for generating a sentence by the grammar rules and domain sentences. Note that the blue tokens and dashed lines are not part of the actual prompt, only shown for illustration.

opt to extract 5 target domain sentences based on preliminary experiments. Since the length of generated sentences is closely related to the number of grammar rules $N$, we use another normal distribution, denoted as $N = \mathcal{N}(N, 3)$ to sample two values, $L_1$ and $L_2$, which define the limits for the length of the generated sentence.

An illustration of the LLMs prompt example is presented in Figure 3, and we utilize gpt-3.5-turbo with the temperature set to 0 for the LLMs generation process.

## 4 Experiments

### 4.1 Data

We use the PTB as our source domain (newswire) and the Multi-domain Constituent TreeBank (MCTB) as the target domain (Yang et al., 2022), covering Dialogue, Forum, Law, Literature, and Review. For validation, we utilize PTB.dev treebank in our cross-domain parsing. For each domain, the vanilla self-training process utilizes the raw corpus of 100k sentences collected from same source as the test set, including Wizard (Dinan et al.), Reddit (Völske et al., 2017), ECtHR (Stiansen and Voeten, 2019), Gutenberg[2], and Amazon (He and McAuley, 2016). This signifies that the source training data and the target test set sentences are homologous, thereby guaranteeing the robustness of the vanilla (i.e., standard) self-training method as our baseline.

To further enhance the quality of the crawled raw corpus, we filter out sentences that are either too long (number of words > 100) or too short (number of words < 3). Then, we sample 40k raw sentences for the Vanilla self-training method. The raw sentence length statistics for the selected 40k and the LLM-generated sentences during the four

---

[2]https://www.gutenberg.org/

| Method | Criteria | Dialogue | Forum | Law | Literature | Review | Avg |
|--------|----------|----------|-------|-----|------------|--------|-----|
| gpt-3.5-turbo | - | $70.70_{*42.4\%}$ | $71.56_{*21.2\%}$ | $80.72_{*27.6\%}$ | $72.83_{*11.4\%}$ | $71.24_{*36.7\%}$ | $73.41_{*27.9\%}$ |
| Kitaev and Klein (2018) | - | 85.92 | 86.00 | 91.71 | 85.04 | 83.51 | 86.44 |
| Vanilla ST | Token | 86.05 | 86.52 | 91.75 | 85.53 | 83.96 | 86.77 |
| | Conf | 86.03 | 86.54 | 91.94 | 85.60 | 83.93 | 86.81 |
| | GRs | 86.04 | 86.49 | 91.92 | 85.63 | 83.94 | 86.80 |
| | GRsConf | 86.13 | 86.57 | 91.93 | 85.76 | 84.02 | 86.88 |
| LLM-enhanced ST | Token | 86.01 | 86.52 | 91.69 | 85.55 | 83.92 | 86.74 |
| | Conf | 86.26 | 86.67 | 92.00 | 85.90 | 84.07 | 86.98 |
| | GRs | 86.18 | 86.82 | 91.97 | 86.00 | 84.09 | 87.01 |
| | GRsConf ‡ | **86.71** | **87.10** | **92.30** | **86.17** | **84.34** | **87.32** |
| Liu and Zhang (2017)* † | - | 85.56 | 86.33 | 91.50 | 84.96 | 83.89 | 86.45 |
| Kitaev and Klein (2018)* † | - | 86.30 | 87.04 | 92.06 | 86.26 | 84.34 | 86.20 |
| LLM-enhanced ST* | GRsConf ‡ | **87.59** | **87.55** | **93.29** | **87.54** | **85.58** | **88.31** |

Table 1: Main results of Vanilla and LLM-enhanced Self-training (ST) with four pseudo-data selection criteria: Token, Conf, GRs and GRsConf. * and † denote the results based on large bert and referred from Yang et al. (2022), respectively. The scaling probability applied to gpt-3.5-turbo results corresponds to the proportion of correct outputs. The ‡ indicates statistical significance compared to baselines with p < 0.05 by paired t-test.

iterations are included in appendix A.1. To better understand the characteristics of our generated sentences, we also provide several typical examples for both crawled and LLM-generated raw sentences in appendix A.2.

## 4.2 Parameters

We employ the same parser for our self-training methods as in Kitaev and Klein (2018)'s work. During the self-training iteration, the raw corpus is initially segmented by Stanza (Qi et al., 2020) and subsequently tagged by the trained parser. The self-training process in all cases comprises 4 iterations and involves selecting the $topK = 2k$ pseudo-data to be integrated as additional training instances for the subsequent iteration. In the vanilla self-training, we choose the topK high-quality instances from a pool of 100k examples and remove the selected sentences from the raw corpus. In contrast, for the LLM-enhanced method, we select the topK data from a pool of 10k examples, as LLMs generate 10k raw sentences for self-training in each iteration.

For the LLM-enhanced constituency parser, we extract grammar rules from current available treebank and integrate them with gpt-3.5-turbo for generating raw corpora. All the parsers employ three distinct seeds, and the performance is measured as the average F1 score.

## 4.3 Main Results

For convenience, the main comparative experiments were conducted using bert-base-uncased, and only the best methods were further experimented on bert-large-uncased. The performance of the constituency parser on five target domains is reported in Table 1.

In the first stage of our experiment, we assessed the performance of gpt-3.5-turbo on few-shot settings for constituency parsing on five target domains. We provided the model with three gold-annotated parse trees paired with sentences from the PTB as demonstrations, and then had gpt-3.5-turbo generate bracketed trees for target domain sentences. However, due to issues such as missing elements and mismatched brackets in LLM's output, more than half of the parse trees are unavailable. For the five target domains under consideration, each comprising 1,000 test samples, the number of available outputs is 424 (Dialogue), 212 (Forum), 276 (Law), 114 (Literature), and 367 (Review), respectively. The LLM exhibits domain bias in the formatting errors of parse tree. It is important to highlight that the reported scores are likely higher than the actual performance, and the scores presented in the main table have been adjusted by multiplying the corresponding available probability. Furthermore, compared to the other domains, gpt-3.5-turbo demonstrates a significantly better performance in constituency parsing for Law domain, just looking at the correctly formatted parsing results.

Secondly, we investigated direct model transfer for cross-domain constituency parsing, a strong baseline method compared with LLMs' parsing.

We trained the parser on the source PTB treebank and directly applied it to the five target domains. From the results, we observed varying distances between the five target domains and the source domain, with the Law domain being closest and the Review domain being farthest in similarity. The difference in F1 scores between Law domain and Review domain is $91.71 - 83.51 = 8.2$ points. On average, the performance of the model transfer method based on bert-base-uncased surpasses that of the Large Language Model's parsing, which is $86.44 - 73.41 = 13.03$.

In the third stage, we examined vanilla self-training using four different selection strategies. From the observation, we find that the optimal selection strategy is not the same for the five target domains. In the Dialogue and Literature domains, the selection based on GRsConf apparently obtained the best performance . We also noticed that the Forum and Review domains exhibit only slight variations across the four pseudo-data selection criteria. However, for the Law domain, employing only the confidence-based criteria is the best choice to achieve self-training improvements. The token-based selection criteria do not demonstrate a significant advantage; they still improved the constituency parser by 0.33 compared to the model transfer. Looking at the average performance, it becomes evident that the selection strategy GRsConf is relatively superior compared to other approaches. This highlights the effectiveness of criteria combination, which not only considers the structural information but also ensures data reliability.

Subsequently, we explored LLM-enhanced self-training for constituency parsers, employing the four selection strategies. The superiority of LLM-enhanced self-training consistency parsing over the vanilla approach is evident across all selection criteria, except for the Token-based selection, where the latter performs better. Furthermore, it is notable that the GRs-based method show a bit more enhancements compared to the Conf-based selection. This highlights that the effectiveness of selection criteria is significantly influenced by the quality of the raw corpus utilized in self-training. The positive results also demonstrate the efficacy of our incremental approach, which uses the LLM to generate target domain sentences in each iteration. Compared to the basic model transfer, our LLM-enhanced method achieves an average improvement of 0.88. The most significant improve-

ment is observed in the Literature domain, while the least improvement is seen in the Law domain. It is worth noting that Yang et al. (2022) used the divergence of grammar rules to measure the distance between different domain constituency parsing treebanks. Among these, the Law domain closely resembles the source domain, exhibiting a minimal improvement of 0.59. Moreover, our LLM-enhanced self-training approach is more effective for domain adaptation tasks with larger difference between the domains.

Additionally, we included two baseline models that employed bert-large-uncased for transition-based and graph-based cross-domain constituency parsing. The results demonstrate that direct model transfer is a relatively effective method. It is important to note that we cannot make a direct comparison with the bert-base-uncased results, as the experimental settings (including seed, batch size, and predict tags) are not entirely consistent.

Lastly, we conducted experiments of the LLM-enhanced self-training method with the best-performing selection strategy **GRsConf** under bert-large-uncased. The approach based on bert-large-uncased outperforms the bert-base-uncased method with anaverage improvement of 0.99. The largest improvement is observed in the Literature domain, with a score increase of $87.54 - 86.17 = 1.37$. On the other hand, the smallest improvement is seen in the Forum domain, with a score increase from $87.55 - 87.10 = 0.45$. These results indicate that utilizing larger pre-trained language models can lead to better performance in the constituency parsing task across various domains.

## 5   Analysis

To conduct a thorough analysis and gain deeper insights into our methods, we have chosen the Review domain for the detailed exploration. Due to space constraints, we placed the comparison between open-source and closed-source LLMs approaches in the appendix A.3

### 5.1   The Instance Selection Strategy

We first investigate four distinct selection strategies for each iteration: Token-based, Conf-based, GRs-based, and GRsConf-based. The line chart in Figure 4 is divided into two partitions, illustrating the parser performance during the iterations for both Vanilla and LLM-enhanced self-training constituency parsing. The chart distinctly shows

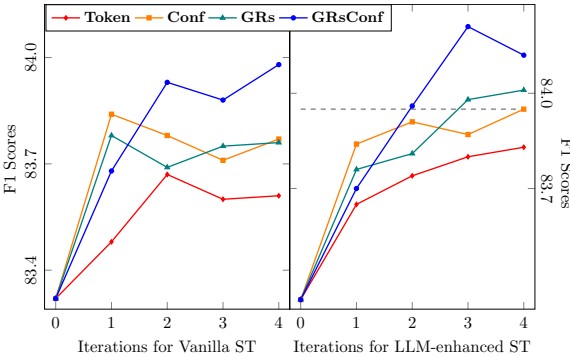

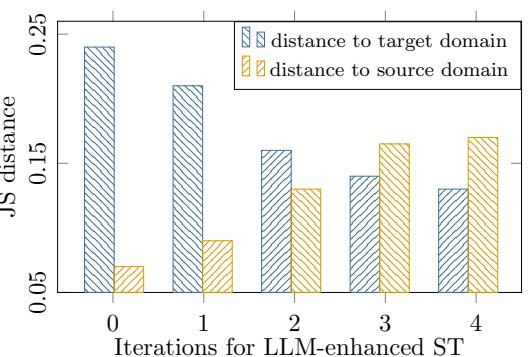

Figure 4: Constituency parsing performance by iteration: Vanilla (left) and LLM-enhanced (right) self-training (ST), the dashed line in the right plot indicates the optimal performance of the Vanilla method.

Figure 5: Distances between Pseudo-data and source and target domains during iterative transfer of Review.

that for the Vanilla method, all strategies except for GRsConf exhibit an initial increase in performance followed by a decrease. This trend suggests that after a few iterations, the candidate data becomes increasingly feature-biased and less suitable for the domain transfer. In the Review domain, the best performance of Vanilla self-training is achieved using with GRsConf-selected pseudo-data.

In contrast, the LLM-enhanced self-training demonstrates a consistent upward trend for all four selection strategies, indicating that the selected data is of high quality and that the adaptation process is both gradual and progressive. This outcome highlights the feasibility and effectiveness of incorporating LLMs into the self-training iteration process, enabling a more fine-grained transfer from the source domain to the target domain. It is also worth noting that the LLM-enhanced method, except for the token-based selection strategy, achieves performance that is either similar to or better than that of the best Vanilla method. The LLM-enhanced method's best performance is achieved using GRsConf, further solidifying the notion that a well-designed selection criterion, when combined with high-quality data, leads to more effective results during the adaptation process.

It is essential to note that our analysis only displays results for four iterations. In our experiments, we also tested up to six iterations. However, the results indicate that, for the Vanilla method, the performance decline becomes increasingly pronounced. And for the LLM-enhanced self-training, no further improvement in performance is observed beyond the fourth iteration.

## 5.2 Pseudo-data from GRsConf

In the context of cross-domain constituency parsing based on LLM-enhanced self-training, the key to performance improvement lies in whether the selected pseudo-data gradually moves closer to the target domain. The LLM generation process and the selection strategies guide the iterations from two opposite directions: the LLM-generated raw text progressively shifts towards the target domain, while the selection criteria aim to ensure that the pseudo-data remains close to the source domain. Consequently, we analyze the best selection strategy for the Review domain, GRsConf, and examine the distribution of the selected pseudo-data during each iteration. Following the work of Yang et al. (2022), we also use the JS divergence of GRs to measure the distance between the selected pseudo-data and both the source and target domains.

As depicted in the chart (Figure 5), the distance between the selected pseudo-data and the source domain increases, while the distance to the target domain gradually decreases. The trend reveals that domain transfer is minimal in the first iteration, with more substantial adaptation occurring in the second and third iterations, and eventually stabilizing in the fourth iteration. This distance evolutionary trends suggests that the domain transfer process is both gradual and progressive, corroborating the effectiveness of the GRsConf selection strategy combined with LLM-enhanced self-training for cross-domain constituency parsing.

## 5.3 Target Sentences Effect

To further investigate the impact of the number of target domain sentences on our LLM-enhanced self-training, we conducted a comparative experiment in the Review domain using the pseudo-data selec-

tion method based on GRsConf. We compared the parser performance, setting the number of target domain sentences to 0, 5, 10, and 20 respectively. As shown in Table 2, we can conclude that the quantity of sentences does not significantly impact the final target domain parser. Moreover, when no target domain sentences were provided and only the target domain name (Review) was given, the results showed a decrease in performance. Further analysis revealed that chatGPT's generated sentences, based on the domain name, significantly differed from the actual domain data.

### 5.4 GRs effect

Additionally, we set up the LLM generation process with 5 target domain sentences, omitting the introduction of grammar rules. According to the experimental results shown in Table 3, it is evident that the parser's performance without the grammar rules is inferior to that of the standard LLM-enhanced self-training approach. This demonstrates that constraining LLM's generation with grammar rules is a reasonable choice.

Regarding grammar rules, there are other intriguing findings to consider. For example, treebanks from different domains exhibit a long-tail distribution of grammar rules. While sharing a considerable number of grammar rules among themselves, each domain possesses a significant count of unique grammar rules, albeit in smaller proportions. Furthermore, our LLM-enhanced self-training method not only modifies the distribution of grammar rules within the training instances but also integrates previously unseen grammar rules.

### 6 Conlusion

In this study, we introduced an innovative LLM-enhanced self-training method for cross-domain adaptation in constituency parsing. By harnessing the generation of LLMs and integrating them into the self-training process, we showed that our approach considerably enhances constituency parsing performance across various domains. Our method effectively merges high-confidence selection cri-

| Model | 0* | 5 | 10 | 20 |
|---|---|---|---|---|
| LLM-enhanced ST | 83.33 | 84.12 | 84.09 | 84.13 |

Table 2: Table 2: Results for various numbers of target sentences. * notes that when no target sentence is provided, we supply the LLM with the domain name.

| Models | GRs | No GRs |
|---|---|---|
| LLM-enhanced ST | 84.12 | 83.15 |

Table 3: Results of using GRs VS no using GRs.

teria with grammar-rule-based selection, progressively moving the training data closer to the target domain. Through experiments, we demonstrated that our method's domain transfer is effective, resulting in improved performance within the target domain. In conclusion, our LLM-enhanced self-training approach offers a promising solution for cross-domain adaptation tasks.

### Ackonwledge

We sincerely thank the reviewers for their invaluable feedback, which significantly improved the quality of this work. We are deeply grateful to the National Natural Science Foundation of China (NSFC) for their funding support, under Grant Nos.61976180 and Nos.62176180, which made this research possible.

### Limitations

because of the expensive cost, we do not employ the GPT-4.It's also intresting that target domain A self-training also improve target B domain constituency parsing performance, which we will explore in the next work. There are many detailed exploration for LLM-equipped self training in the raw corpus generation partition, e.g. the influence of the prompt write by different people . For time constraints, we just chose a falcon-40-instruct, and the smaller LLMs are also worth a try.

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

| Category | Dialogue | Forum | Law | Literature | Review |
|---|---|---|---|---|---|
| Test Set | 13.51 | 22.01 | 22.59 | 23.24 | 13.30 |
| Raw Corpus (40k) | 18.67 | 20.72 | 21.74 | 20.30 | 20.56 |
| LLM-itr1 | 23.98 | 37.08 | 39.39 | 37.02 | 22.91 |
| LLM-itr2 (Token) | 23.98 | 37.02 | 39.51 | 37.05 | 23.06 |
| LLM-itr2 (Conf) | 23.97 | 36.97 | 39.31 | 37.07 | 22.95 |
| LLM-itr2 (GRs) | 24.08 | 36.99 | 39.28 | 37.15 | 22.97 |
| LLM-itr2 (GRsConf) | 23.82 | 36.82 | 38.98 | 36.82 | 23.07 |
| LLM-itr3 (Token) | 24.07 | 36.89 | 39.27 | 36.88 | 23.07 |
| LLM-itr3 (Conf) | 24.14 | 36.80 | 38.94 | 36.67 | 22.92 |
| LLM-itr3 (GRs) | 24.01 | 36.93 | 39.21 | 37.08 | 22.98 |
| LLM-itr3 (GRsConf) | 23.99 | 36.88 | 39.26 | 37.01 | 23.03 |
| LLM-itr4 (Token) | 23.95 | 35.78 | 39.26 | 36.95 | 23.09 |
| LLM-itr4 (Conf) | 23.96 | 36.83 | 38.96 | 36.81 | 22.90 |
| LLM-itr4 (GRs) | 24.03 | 36.87 | 39.06 | 37.12 | 22.99 |
| LLM-itr4 (GRsConf) | 23.91 | 36.85 | 38.90 | 37.01 | 23.02 |

Table 4: Length statistics for the test set, crawled raw corpus (40k), and LLM-generated raw sentences for each selection strategy (10k per iteration).

## A Appendix

### A.1 Length of Instances

The length statistics for the test set, crawled raw corpus(40k), and LLM-generate raw sentences for each selection strategy (10k on each iteration) are reported at table 4. It is evident that, from the perspective of sentence length, the raw corpus used for Vanilla self-training is closer to the target domain data compared to the sentences generated by LLM. Interestingly, different selection criteria can affect the average length of the generated sentences. For example, when selecting data based on grammar rules (GRs), the generated sentences tend to be longer, which can be attributed to the fact that the GRs-based criteria prefer sentences contain more complex structures.

### A.2 LLM Generated Sentences

We have selected several crawled raw sentences from domain Review GRsConf selection strategy, which are concise, readable, and more colloquial. In our improved self-training method, LLM generates raw corpora for each iteration, as shown in Figure 6. These sentences incorporate more diverse information and exhibit more complex structures; however, there may be instances where the semantics are not that smooth or coherent.

### A.3 Open Source LLM Exploration

Obtaining raw corpus for each iteration using closed-source LLMs, such as chatGPT, can be time-consuming and costly. Therefore, we investigate the possibility of employing open-source Large Language Models to achieve similar effects. To test

**Crawed Raw Corpus**

1. I would not recommend . 2. I like that the fittings are metal and seem to be very sturdy . 3. Then he learns , she was trying to get to her child who is in a babyseat in the back . 4. Decent movie worth a rental and if you are a collector buy it . 5. ......

**LLM-generated Raw Corpus**

itr 1: 1. The turns of play until the end of the game were intense . 2. All instructions of how to use the product will be given in the manual . 3. ......

itr 2: 1. She is a major client , and I removed the drive from the enclosure to see how warm it was . 2. My hand spelled " I love you " in sign language , but she did n't notice . 3. ......

itr 3: 1. The movie was comfortable , but for corporatism , Mama recommended Telephone in February . 2. I have a disc that has been dented during delivery , and now I need to return it . 3. ......

itr 4: 1. Very old Spiderman is under the CD cases , and it 's easy to place others on top . 2. I really prefer the normal ones , not including those with too many features . 3. ......

Figure 6: Comparison of typical examples of crawled and iterative LLM-generated raw corpus.

| LLM | OpenSrc | R | P | F |
|---|---|---|---|---|
| chatGPT | ✗ | 82.07 | 86.26 | 84.12 |
| falcon | ✓ | 82.04 | 86.16 | 84.05 |

Table 5: Results of chatGPT and falcon. OpenSrc denotes whether the LLM is open source.

this idea, we use the powerful falcon-40b-instruct (Almazrouei et al., 2023) as an alternative open-source LLM in the Review domain based on the GRsConf selection strategy.

In comparison to API requests for chatGPT, the open-source LLM features much faster inference speeds. The experimental results, as shown in Table 5, reveal that using chatGPT's prompt to generate raw text with Falcon results in strong performance when applied to our iterative method. This finding indicates that incorporating open-source LLMs into our self-training approach is also feasible, and the open-source LLM can seamlessly embed into the process in a completely closed-loop manner. This eliminates the need for manual intervention in the iteration process, as is required with chatGPT, making the approach more efficient and cost-effective.