# OpenReview forum: "LLM-enhanced Self-training for Cross-domain Constituency Parsing"
_EMNLP/2023/Conference — EMNLP 2023 Main_

### Official Review · Reviewer_imit · 2023-07-19

**Soundness:** 3

**Excitement:**

4: Strong: This paper deepens the understanding of some phenomenon or lowers the barriers to an existing research direction.

**Paper Topic And Main Contributions:**

The paper presents a novel way to support open-domain transfer for consistency parsing by combining the famous self-training technique with the help of LLMs. The authors propose to generate target domain examples using an LLM given a prompt that contains their grammar rules and raw examples of target sentences. The generated samples with their corresponding trees are then added to the training set and the parser is trained iteratively for 4 iterations.

The authors also propose new methods for self-training data selection using similarity of token-based distribution and similarity of grammar rules and combine them with the known confidence-based method.

Results show competitive but not superior performance of their methods, which weakens the overall claim.

The authors also demonstrate their ability to seamlessly integrate their method with open-source LLMs such as Falcon-40B.

**Questions For The Authors:**

Lines 351-354 I did not quite understand how you sample the grammar rules from PTB. How do you derive the parameters?

**Reasons To Accept:**

Overall, I really like the proposed method. It is quite simple and effective and marries together the timelessly self-training technique with the novel capabilities of LLMs. The paper also nicely combines the training of standard classifiers with pseudo-generated samples. In general, I would like to see more papers of this kind that find novel ways to extend standard techniques with LLMs.

The newly proposed selection methods are also refreshing but don't seem to contribute much to overall performance.

Figures 4 and 5 hold informative analyses and highlight the strengths of the proposed method over baselines.

**Reasons To Reject:**

The proposed approach assumes that the source and target samples obtain similar grammar rules, but it would fail on target domains that obtain different grammar rules, not to mention, sentences from other languages. This is a noticeable disadvantage.

The thing that bothers me the most, if I understand correctly, the authors did not add the same amount of new training samples at each iteration - for vanilla self-training they added 2K samples, while for their method they added 10K samples. This is not a fair comparison and severely harms the results. It isn't clear if the authors then select 2K samples out of the total of 10K generated samples or not. This affects both Table 1 and Figure 4.

Secondly, the authors choose to test all models on the MCTB dataset which only consist of test samples. For supplying unsupervised samples for the vanilla self-training they turned to other "homologous" resources.
However, it is very common, even for open-domains, to have a decent amount of available unlabeled samples. In most cases, self-training directly operates over unlabeled target samples (which do not appear in the held-out test set). I would like to see experiments on such a scenario based on a target dataset that was manually split to train and test samples. That way, the authors could test their method against the standard self-training scheme.

Other than that I find section 5.3 and table 2 to be redundant. The authors could have used an open-source LLM in advance instead or in addition to the reported results on GPT3.5-turbo in Table 1. Or, they could have put it in the appendix. I believe the occupied space could have been used better for a more insightful analysis.

**Reproducibility:**

3: Could reproduce the results with some difficulty. The settings of parameters are underspecified or subjectively determined; the training/evaluation data are not widely available.

**Reviewer Confidence:**

4: Quite sure. I tried to check the important points carefully. It's unlikely, though conceivable, that I missed something that should affect my ratings.

**Typos Grammar Style And Presentation Improvements:**

line 250: the abbreviation of GRs is not yet defined.

line 256: D_hat = \empty_set.

line 421 - chatGPT (overall it is very confusing that you use both chatGPT and GPT3.5-turbo for referring to the same model)

Table 1 - please bold the highest numbers in each column to ease the reading.

line 489 - The most...

---

> ### Author Rebuttal · Authors · 2023-08-29
>
> We appreciate your valuable feedback.
>
> Q0: Not Superior Performance
> it's important to highlight that self-training inherently serves as a strong baseline for cross-domain tasks. Additionally, we've conducted a statistical significance analysis that yielded results with p<0.05, reinforcing our claim that our LLM-enhanced self-training surpasses the strong baseline (vanilla self-training). We will revise and underscore this aspect in the revised manuscript.
>
> Q1: Noticeable Disadvantage for Different Grammar Rules
> We've examined the grammar rules distribution of the MCTB dataset compared with source PTB train set and found that different grammar rules exist in all the domains. However, our method can predict the grammar rules never seen in the training set and self-training transfers gradually grammar-rule distribution from the source domain to the target domain. We will provide a more comprehensive analysis of different grammar rules in the revised manuscript.
>
> Q2: 2K for Vanilla and 10k for Our Method
> For all methods, vanilla self-triaining and LLM-enhanced self-training, we added 2k pseudo trees from a pool of 10k pseudo data. We will clarify this setup further in the camera-ready version.
>
> Q3: Confusion on Standard Self-training Scheme
> In fact, what we referred to as "vanilla self-training" aligns with what you mentioned as the "standard self-training scheme." For the vanilla self-training, unsupervised samples are from the same "homologous" resources not the other "homologous" resources. The sentences of MCTB are selected from the 5 resources (line 377~382) which also served as our homologous resources. We will make this aspect clearer in future version.
>
> Q4: Redundant Open-source LLM in 5.3
> We will relocate Section 5.3 along with Table 2 to the appendix. The comparison experiments involving open-source LLM are limited to one domain due to resource constraints, which couldn't be included directly in Table 1 encompassing five domains. This adjustment in space will provide room for more insightful analysis, as you suggested.
>
> Q5: Typos Grammar Style And Presentation Improvements
> In the revised version, we will improve all the writing and stylistic issues you've raised for improvement.

---

### Official Review · Reviewer_wYfy · 2023-08-03

**Soundness:** 4

**Excitement:**

3: Ambivalent: It has merits (e.g., it reports state-of-the-art results, the idea is nice), but there are key weaknesses (e.g., it describes incremental work), and it can significantly benefit from another round of revision. However, I won't object to accepting it if my co-reviewers champion it.

**Paper Topic And Main Contributions:**

The authors propose a LLM-enhanced self-training for constituency parsing on different domains. Using causal LLMs, they propose an iterative method consisting in generating new training instances on a target domain using grammar-rules extracted from a source domain treebank and few samples on the target domain. Then they select the synthetic samples with higher confidence and update the source treebank with these silver data, and udpate the grammar rules with the updated treebank.

Overall, the proposed approach is interesting and obtains better results than the baselines. However, experimental setup is a bit weak. Although they perform a detailed discussion on results, thee improvements are small and are not controlled for seed randomization or tested their statistical significance, so the claims they made are not really supported.

**Questions For The Authors:**

1. How exactly are the grammar rules extracted? I am not sure I fully grasp the method.
2. In lines 385, 386, lenght refers to tokens? It should be specified.

**Reasons To Accept:**

1. Method and discussion of results are described in detail.
2. Simple and elegath proposed method.
3. Comparison of open- and closed-source language models.

**Reasons To Reject:**

1. Weak experimental setups. Improvements are small and without a statistical significance analysis, it is difficult to extract conclusions from them.

**Reproducibility:**

4: Could mostly reproduce the results, but there may be some variation because of sample variance or minor variations in their interpretation of the protocol or method.

**Reviewer Confidence:**

2: Willing to defend my evaluation, but it is fairly likely that I missed some details, didn't understand some central points, or can't be sure about the novelty of the work.

---

> ### Author Rebuttal · Authors · 2023-08-29
>
> Thanks for the insightful comments.
>
> Q0: Weak Experimental Setup and Small Improvements
> We have already performed an statistical significance analysis where p<0.05 on three different random seed. The results support our claim that our LLM-enhanced self-training can outperform the strong baseline (vanilla self-traning). We will update the results in the revised manuscript.
>
> Q1: Grammar Rules Extraction
> The grammar rules are directly extracted from the constituent tree, where the parent node corresponds to the left hand of the grammar rule, and all child nodes correspond to the right tail side. In the camera-ready version, we will ensure that readers can fully understand the extraction method without ambiguity.
>
> Q2: Clarification on "length" in Lines 385-386
> Thank you for pointing out the need for clarity regarding the term "length" in Lines 385-386. In the revised version, we will specify that "length" refers to the number of tokens, providing readers with precise information to better understand our approach.

---

### Official Review · Reviewer_8ZMA · 2023-08-05

**Soundness:** 3

**Excitement:**

3: Ambivalent: It has merits (e.g., it reports state-of-the-art results, the idea is nice), but there are key weaknesses (e.g., it describes incremental work), and it can significantly benefit from another round of revision. However, I won't object to accepting it if my co-reviewers champion it.

**Paper Topic And Main Contributions:**

This paper explores the use of LLM to create target-domain text for improving constituency parsing in a cross-domain self-training setting. The approach involves an iterative process where an initial baseline constituency parser (berkely-neural-parser) is trained on a source domain. LLM is then prompted with grammar rules from the current set of source+pseudo trees and sample sentences from the target domain to generate new text in the target domain. A selection process with certain criteria is used to choose 'high-quality' samples as pseudo training data for the next round of parser training. The experiments conducted on the PTB->MCTB cross-domain setting demonstrate that the proposed LLM-based self-training approach achieves slightly better performance (0.1~0.2 improvement in F1 score) compared to a vanilla ST approach that has access to a small fixed set of target text. Some analysis shows that the LLM-based approach consistently improves during the iteration process, and the selected pseudo text gradually becomes more similar to the target domain.

**Reasons To Accept:**

The investigation of using LLM to generate target domain text for domain adaptation in the self-training setting is an interesting problem, and the paper demonstrates some positive results despite the small improvement.

**Reasons To Reject:**

- Since the proposed LLM-based approach aims to benefit from low-resource target domains, it is necessary to evaluate how the size of available target text affects performance. However, the paper only investigates one fixed amount of target text, leaving important questions unanswered. For example, does the amount of available target text affect the diversity of the generated text and impact the effectiveness of the proposed approach? Would this approach be more effective if you further reduce the amount of target text?
- The paper emphasizes using grammar rules as prompts for LLM to generate target-domain text, but no experiments were conducted to evaluate the benefits it brings.
- The writing could be improved to provide more implementation details. For instance, the hyperparameters used for constructing LLM prompts in section 3.5 are not detailed. The implementation details of the instance selection criteria are not provided, such as the confidence measure and how it is combined with the grammar-rule-based selection approach.

**Reproducibility:**

3: Could reproduce the results with some difficulty. The settings of parameters are underspecified or subjectively determined; the training/evaluation data are not widely available.

**Reviewer Confidence:**

4: Quite sure. I tried to check the important points carefully. It's unlikely, though conceivable, that I missed something that should affect my ratings.

---

> ### Author Rebuttal · Authors · 2023-08-29
>
> We sincerely appreciate your insightful comments.
>
> Q0: Evaluation for Available Target Text Size
> In the pre-experiment, we simply tested performance using 5, 10, and 20 available target texts. There are no significant differences on these three settings. Considering resource limitations, we select 5 samples for efficiency. In the final version, we will conduct a more extensive analysis, delving into the influence of different target text quantities on LLM-generated text, including diversity.
>
> Q1: Evaluation for Grammar Rules Effect
> In the revised manuscript, we will dedicate a section to conducting experiments that exclude grammar rules from the prompt generation process and examine the effect of grammar rules in our method.
>
> Q2:  Improved Implementation Detail
> We will refine the implementation details in the camera-ready version. The LLM parameter information will be relocated from Section 4.3 to Section 3.5 for clearer presentation. We will improve the details in demonstrating the instance selection criteria and ensure its clarity in future version.

---

### Meta-Review · Area_Chair_6D92 · 2023-09-18

**Recommendation:** 5

**Metareview:**

The paper presents an interesting approach to improve cross-domain parsing, with a clever way to use LLMs, producing small gains over reasonable baselines. The reviews found the use of LLMs to produce target domain text for self-training to be an appealing idea, and suggested several ways in which the paper could be strengthened, including the inclusion of results over multiple runs with random seeds, and clarification of details such as whether the size of available target texts affects performance, whether different data amounts are used in vanilla vs LLM-enhanced self-training, etc. The responses provided by the authors seem satisfactory.

---

### Decision · Program_Chairs · 2023-10-07

**Decision:**

Accept-Main

**Comment:**

The paper presents an interesting approach to improve cross-domain parsing, with a clever way to use LLMs, producing small gains over reasonable baselines. The reviews found the use of LLMs to produce target domain text for self-training to be an appealing idea, and suggested several ways in which the paper could be strengthened, including the inclusion of results over multiple runs with random seeds, and clarification of details such as whether the size of available target texts affects performance, whether different data amounts are used in vanilla vs LLM-enhanced self-training, etc. The responses provided by the authors seem satisfactory.